Predicting no-shows at outpatient appointments in internal medicine using machine learning models

Ocampo Osorio Felipe 1 2 3 4
Pedroza Gomez Santiago 1 2
http://orcid.org/0000-0002-9680-9108 Rebellón Sanchez David Esteban 1 2
Ramirez Fernandez Richard 1 2
Tabares-Soto Reinel 3 5
Bravo-Ortíz Mario Alejandro 3 5 6
Cruz Suarez Gustavo Adolfo 1 2 4 gustavo.cruz@fvl.org.co
1 Unidad de Inteligencia Artificial, Fundación Valle del Lili , Cali, Valle del Cauca , Colombia
2 Centro de Investigaciones Clínicas, Fundación Valle del Lili , Cali, Valle del Cauca , Colombia
3 Departamento de Electrónica y Automatización, Universidad Autónoma de Manizales , Manizales, Caldas , Colombia
4 Departamento de Salud Pública y Medicina Comunitaria, Universidad ICESI , Cali, Valle del Cauca , Colombia
5 Departamento de Sistemas e Informática, Universidad de Caldas , Manizales, Caldas , Colombia
6 Centro de Bioinformática y Biología Computacional (BIOS) , Manizales, Caldas , Colombia
Zhou Jiayan
Electronic publication date: 2025 Apr 22
Publication date: 2025
Volume: 11
Electronic Location ID: e2762
Received 2024 Nov 13; Accepted 2025 Feb 21
Copyright: © 2025 Ocampo Osorio et al.
Copyright year: 2025
Copyright holder: Ocampo Osorio et al.
License: This is an open access article distributed under the terms of the Creative Commons Attribution License, which permits unrestricted use, distribution, reproduction and adaptation in any medium and for any purpose provided that it is properly attributed. For attribution, the original author(s), title, publication source (PeerJ Computer Science) and either DOI or URL of the article must be cited.
License URL: https://creativecommons.org/licenses/by/4.0/

Keywords: Internal medicine, Machine learning, Medical appointments, Non-attendance, No-shows

Funding: The authors received no funding for this work.

==============================
The high prevalence of patient absenteeism in medical appointments poses significant challenges for healthcare providers and patients, causing delays in service delivery and increasing operational inefficiencies. Addressing this issue is crucial in the internal medicine department, a fundamental pillar of comprehensive adult healthcare that manages various chronic and complex conditions. To mitigate absenteeism, we present an innovative application of machine learning models specifically designed to predict the risk of patient absenteeism in the internal medicine department of Fundación Valle del Lili, a high-complexity hospital in Colombia. Leveraging an institutional database, we conducted a statistical analysis to identify critical variables influencing absenteeism risk, including clinical and sociodemographic factors and characteristics of previously attended appointments. Our study evaluated seven distinct machine learning models, explored various data processing techniques, and addressed class imbalance through oversampling and undersampling strategies. Hyperparameter optimization was conducted for each model configuration, culminating in selecting the Bagging RandomForest model, which demonstrated outstanding performance when combined with standardized data and balanced using the Synthetic Minority Oversampling Technique (SMOTE). Additionally, Shapley values (SHAP) were applied to enhance the interpretability of the model, enabling the identification of the most influential variables in predicting medical absenteeism, such as the number of previous absences, the day and month of the appointment, and diagnosed diseases. The selected model achieved a predictive accuracy of 84.80 ± 0.81%, an AUC value of 0.89, an F1-score of 84.75%, and a recall of 83.02% in cross-validation experiments. These results highlight the potential of our experimental approach to identify the most suitable model for proactively predicting patients at high risk of absenteeism, optimizing resource allocation, and improving the quality of medical care in internal medicine in the future. Our methodology provides a foundation for reducing operational inefficiencies and strengthening intervention strategies. This benefits healthcare providers and patients through more timely and effective care. Ultimately, this approach contributes to improving patient outcomes and institutional efficiency.

Introduction

The non-attendance to medical appointments is considered highly relevant due to its multifaceted impact on the healthcare system. Firstly, this issue directly affects healthcare by causing delays in diagnosing and treating diseases, which can negatively affect patients’ health. Additionally, missing medical appointments represents a significant waste of medical resources, including healthcare staff time, medical facilities, and operational costs, affecting the efficiency and economy of the healthcare system (Triemstra & Lowery, 2018). The continuity of care is also compromised, especially for patients with chronic diseases, as regular planned care is disrupted. At the access level, this issue can hinder the availability of appointments for other patients, increasing waiting lists and impeding timely access to care, thus generating a significant economic impact on the healthcare system and patients by accumulating opportunity costs and their associated expenses (Loccoh et al., 2018). This phenomenon is a global concern. In 2018, a global meta-analysis of the rate of missed medical appointments was conducted. On average, the no-show rate was 23%, ranging from 4% to 79% depending on the treating medical specialty and the type of appointment (Dantas et al., 2018). To address and reduce the problem of missed medical appointments, hospitals have adopted multiple strategies, including sending appointment reminders through email, phone calls, and text messages (Liew et al., 2009). These strategies aim to improve the accessibility of healthcare and ultimately reduce missed medical appointments. However, despite multiple efforts and strategies implemented by healthcare institutions, the patient no-show rate continues to rise, posing challenges to healthcare service delivery.

This work’s main contribution is identifying a computational model to predict absences in outpatient appointments in the internal medicine department at Fundación Valle del Lili, a high-complexity hospital located in Colombia. The internal medicine department was chosen for its fundamental role in providing comprehensive and specialized medical care to adult patients, addressing a wide range of chronic and complex conditions requiring continuous and personalized attention. Additionally, this area has one of the highest patient flows within the institution, making it an ideal environment to implement and evaluate a computational model to predict absences in outpatient appointments. For this purpose, an institutional database containing historical records of patients treated in this specialty was used. This database includes clinical and sociodemographic characteristics and details of the last scheduled appointment, whose influence on absenteeism was validated by identifying p and q values. A data combination process was carried out during preprocessing to validate the computational model’s effectiveness.

The decision to address the problem using machine learning models is based on the structure of the available data, allowing for the analysis of complex and nonlinear relationships inherent to the clinical context. The Bagging RandomForest model, combined with the Synthetic Minority Oversampling Technique (SMOTE) oversampling technique, was selected for its high accuracy and performance in identifying absenteeism patterns, improving the representation of minority cases without introducing significant bias. Additionally, SHapley Additive exPlanations (SHAP) values were employed to enhance the interpretability of the model, identifying the most influential variables such as the number of previous absences, the day and month of the appointment, and diagnosed diseases. This approach not only strengthened the model’s accuracy but also provided key insights for developing strategies aimed at patients with a higher risk of absenteeism, ensuring a robust, transparent, and practical analysis for managing medical absenteeism. The results of this research provide a starting point for optimizing resource management and improving efficiency in the healthcare sector by developing strategies based on identifying users at high risk of missing their appointments.

Related work

Artificial intelligence techniques are now being used to identify patterns in medical non-attendance and detect patients at risk of missing appointments. These techniques include statistical analysis, machine learning, and deep learning, which can help develop programs to address these gaps. As a result, there is growing research and innovation in this field, with the development of applications that use artificial intelligence to identify patterns and patients prone to non-attendance and apply interventions to reduce this rate. The results of these efforts are highlighted below: Harvey and colleagues conducted radiology absence prediction using regression models with data extracted from electronic medical records. They recorded demographic, clinical, and health service use variables relevant to attendance. Descriptive statistics and logistic regression models were applied to determine whether these data could predict missed scheduled radiology exams. The regression models demonstrated predictive accuracy with an area under the curve of 0.754. The factors most strongly predicted absence were previous absences, days between scheduling and appointment, modality, and insurance type (Harvey et al., 2017). Chua & Chow (2019) developed a predictive model to stratify the risk of nonattendance in specialized outpatient clinics in a public hospital and improve efficiency in the use of resources. Univariate analyses were performed on 16 variables, including demographic data, appointment/visit records, and historical outpatient records. Multiple logistic regression was used to identify independent risk factors for no-shows. From univariate analysis, 11 variables were determined to be associated with no-shows, and six of these (age, race, specialty, delivery time, referral source, and prior visit status) remained independently associated with the model. The model’s predictive ability was assessed by receiver operating curve analysis, with an area under the curve of 72% (Chua & Chow, 2019).

Daghistani et al. (2020) applied predictors of outpatient absence through extensive data analysis using Apache Spark. After conducting several experiments and employing validation methods, it was found that gradient boosting (GB) performed the best, with an increase in accuracy and ROC curve to 79% and 81%, respectively. This study highlights the importance of exploring and evaluating the performance of machine learning models using multiple evaluation methods, as prediction accuracy can vary significantly (Daghistani et al., 2020). Nelson et al. (2019) evaluated complex machine learning models for predicting scheduled hospital attendance with artificial intelligence. They used models based on logistic regression, support vector machines, random forest, AdaBoost, and gradient boost machines, varying in complexity. The high-dimensional gradient boosting machine models performed best, with an receiver operating characteristic (ROC) value of 0.852 and an average accuracy of 0.511. To achieve this performance, 81 variables were required. This study highlights the need for more complex models to predict optimal attendance, reflecting the interaction of multiple patient, environmental, and operational causal factors (Nelson et al., 2019).

The results of these and other research studies not mentioned in this section provide ample information on using artificial intelligence techniques to predict patient absenteeism to medical appointments. Thus, there is a need to evaluate the computational parameters necessary for these algorithms to reduce the incidence rates of non-attendance and implement actions that subsequently support the shortening of this phenomenon. Therefore, it is necessary to parameterize the design and implementation processes of the computational models, in addition to identifying those critical factors that contribute to the increase of no-shows.

Materials and Methods

Database

The primary source of information for data collection was the electronic medical records stored in SAP Logon 750, the institution’s medical records storage software. This source structured historical data of patients treated in the internal medicine specialty between January 2019 and September 2022. The variables included sociodemographic information, clinical conditions, and characteristics of the consultations, resulting in a total of 13 features, as show in Table 1.

Table 1 Sociodemographic, clinical, and medical appointment characteristics.

Variable Name	Definition	Type of variable and operational level	
Sociodemographic characteristics	
Age	Age in years	Quantitative, continuous (Integer)	
Sex	Patient gender	Qualitative, nominal (0: Female, 1: Male, 2: Undetermined)	
Insurance type	Patient affiliation	Qualitative, nominal (0: ARL, 1: Contributory, 2: Subsidized, 3: Prepaid, 4: Private, 5: International Private, 6: Other, 7: Other Institutional, 8: Prepaid International)	
Clinical characteristics	
Number of diseases	Number of patient diagnoses	Quantitative, ordinal (Integer)	
Recent hospitalization	Number of hospitalizations in last 3 months	Quantitative, ordinal (Integer)	
Number of medications	Number of medications used per day	Quantitative, ordinal (Integer)	
Medical appointment characteristics	
Hour	Appointment time	Quantitative, continuous (Integer)	
Day	Appointment day	Qualitative, nominal (0: Monday, 1: Tuesday, 2: Wednesday, 3: Thursday, 4: Friday, 5: Saturday, 6: Sunday)	
Month	Appointment month	Qualitative, nominal (1: January, 2: February, 3: March, 4: April, 5: May, 6: June, 7: July, 8: August, 9: September, 10: October, 11: November, 12: December)	
Creation to assignment interval	Days between consultation order and appointment day	Quantitative, continuous (Integer)	
Number of previous attendance	Number of prior consultations (Jan 2021–Sept 2022)	Quantitative, ordinal (Integer)	
Number of previous non-attendance	Number of previous non-attendances (Jan 2021–Sept 2022)	Quantitative, ordinal (Integer)	
Appointment type	Attendance/Absence at last internal medicine appointment	Qualitative, nominal (0: Absence, 1: Attendance)	

The study variables were selected based on an in-depth analysis of existing scientific literature, specific characteristics of the study population, and recommendations from an expert panel in the outpatient unit (Dantas et al., 2018; Harvey et al., 2017; Hernández-García et al., 2018; Salah & Srinivas, 2022).

Data distribution

Historical data were collected from 18,587 patients who attended the Internal Medicine specialty between January 2019 and September 2022. Of this total, 12,153 patients attended scheduled medical appointments, while 6,434 recorded absences. The distribution of these figures is shown in Fig. 1A. It is evident that there is an imbalance in the classes, which could negatively impact the performance of computational models, biasing them towards the majority class.

Figure 1 Data distribution: (A) Distribution of the data referring to the database, and (B) age distribution dataset by class.

In the exploration of the database, a variation was observed in the distributions of ages related to the outcome of the last medical appointment, revealing notable differences between different demographic groups. In particular, it was found that older adults show a lower prevalence of missed medical appointments compared to other groups (Fig. 1B). These findings highlight the complexity of factors affecting medical appointment adherence and emphasize the importance of considering multiple variables in healthcare analyses.

Data statistics

In this study, we analyzed the data to elucidate the factors that influence non-attendance at scheduled internal medicine consultations. Using the database provided, we examined several variables and applied statistical methods to identify significant patterns and relationships. We wanted to provide a deeper understanding of these factors to offer practical recommendations to healthcare professionals to improve patient’s adherence to their medical appointments. Quantitative variables were evaluated using t-tests or Mann-Whitney U-tests according to their normality distribution, while categorical variables were evaluated using Chi-square tests. The p-values were subsequently adjusted using the Benjamini-Hochberg correction method, generating q-values, thus constructing a comparative table, as shown in Table 2.

Table 2 Patient characteristics.

Characteristic	Overall	Absences	Attendances	p-value	q-value	
	N = 18,587	N = 6,434	N = 12,153			
Age				<0.001	<0.001	
Mean (SD)	50.02 (17.31)	47.32 (17.52)	51.45 (17.03)			
Median (IQR)	50 (37, 62)	46 (34, 59)	52 (38, 63)			
Sex				0.2	0.2	
Female	11,505 (62.19%)	3,903 (61.49%)	7,602 (62.55%)			
Male	6,995 (37.81%)	2,444 (38.51%)	4,551 (37.45%)			
Unknown	87	87	0			
Insurance type				<0.001	<0.001	
0: ARL	602 (3.24%)	101 (1.57%)	501 (4.12%)			
1: Contributory	1,685 (9.07%)	298 (4.63%)	1,387 (11.41%)			
2: Subsidized	43 (0.23%)	11 (0.17%)	32 (0.26%)			
3: Prepaid	15,411 (82.91%)	5,659 (87.95%)	9,752 (80.24%)			
4: Private	667 (3.59%)	321 (4.99%)	346 (2.85%)			
5: International private	13 (0.07%)	4 (0.06%)	9 (0.07%)			
6: Other	7 (0.04%)	4 (0.06%)	3 (0.02%)			
7: Other institutional	153 (0.82%)	35 (0.54%)	118 (0.97%)			
8: Prepaid international	6 (0.03%)	1 (0.02%)	5 (0.04%)			
Number of diseases				<0.001	<0.001	
Mean (SD)	1.44 (1.11)	1.23 (1.11)	1.55 (1.10)			
Median (IQR)	1 (1, 2)	1 (1, 2)	1 (1, 2)			
Recent hospitalization				0.5	0.5	
Mean (SD)	0.04 (0.21)	0.04 (0.24)	0.04 (0.19)			
Median (IQR)	0 (0, 0)	0 (0, 0)	0 (0, 0)			
Number of medications				<0.001	<0.001	
Mean (SD)	0.54 (1.23)	0.32 (0.93)	0.66 (1.35)			
Median (IQR)	0 (0, 0)	0 (0, 0)	0 (0, 1)			
Hour of appointment				<0.001	<0.001	
6:00 to 8:00	11 (0.12%)	3 (0.10%)	8 (0.13%)			
8:00 to 10:00	2,033 (22.02%)	720 (22.86%)	1,313 (21.58%)			
10:00 to 12:00	2,435 (26.38%)	749 (23.79%)	1,686 (27.72%)			
12:00 to 14:00	717 (7.77%)	181 (5.75%)	536 (8.81%)			
14:00 to 16:00	2,266 (24.55%)	824 (26.17%)	1,442 (23.71%)			
16:00 to 18:00	1,770 (19.17%)	672 (21.34%)	1,098 (18.05%)			
18:00 to 20:00	9,355	3,285	6,070			
Day of the appointment				<0.001	<0.001	
Sunday	3,294 (17.85%)	1,224 (19.41%)	2,070 (17.03%)			
Monday	2,869 (15.54%)	1,246 (19.76%)	1,623 (13.35%)			
Tuesday	2,643 (14.32%)	1,129 (17.91%)	1,514 (12.46%)			
Wednesday	2,814 (15.25%)	1,148 (18.21%)	1,666 (13.71%)			
Thursday	2,327 (12.61%)	841 (13.34%)	1,486 (12.23%)			
Friday	2,013 (10.91%)	348 (5.52%)	1,665 (13.70%)			
Saturday	2,498 (13.53%)	369 (5.85%)	2,129 (17.52%)			
Month of the appointment				<0.001	<0.001	
January	1,225 (6.60%)	360 (5.63%)	865 (7.12%)			
February	1,138 (6.13%)	401 (6.27%)	737 (6.06%)			
March	1,418 (7.64%)	488 (7.63%)	930 (7.65%)			
April	1,431 (7.71%)	591 (9.24%)	840 (6.91%)			
May	1,750 (9.43%)	574 (8.97%)	1,176 (9.68%)			
June	1,949 (10.51%)	590 (9.22%)	1,359 (11.18%)			
July	2,212 (11.92%)	553 (8.64%)	1,659 (13.65%)			
August	3,070 (16.55%)	838 (13.10%)	2,232 (18.37%)			
September	3,182 (17.15%)	827 (12.93%)	2,355 (19.38%)			
October	378 (2.04%)	378 (5.91%)	0 (0%)			
November	407 (2.19%)	407 (6.36%)	0 (0%)			
December	391 (2.11%)	391 (6.11%)	0 (0%)			
Creation to assignment interval				0.007	0.009	
Mean (SD)	25.60 (28.66)	25.59 (25.35)	25.61 (30.26)			
Median (IQR)	19 (8, 35)	20 (8, 34)	19 (7, 35)			
Number of previous attendances				<0.001	<0.001	
Mean (SD)	2.68 (2.26)	2.69 (2.50)	2.68 (2.12)			
Median (IQR)	2 (1, 4)	2 (1, 4)	2 (1, 4)			
Number of previous non- attendance				<0.001	<0.001	
Mean (SD)	2.04 (3.43)	2.99 (3.96)	1.54 (3)			
Median (IQR)	1 (0, 3)	2 (1, 4)	1 (0, 2)			

Age, insurance type, number of diseases, medications, day and month of appointment assignment, and prior attendance and missed appointments were significant factors associated with absenteeism (p < 0.001, q < 0.001). Most participants in both cohorts had prepaid insurance coverage (>80%). Specifically, those associated with prepaid insurance exhibited a high absenteeism rate of 87.95%. It was observed that Monday was the day with the highest medical appointment assignment (15.54%), coinciding with the highest absenteeism rate of 19.76%. Likewise, August stood out as the month with the highest appointment assignment (16.55%), with an absenteeism rate of 13.10%. As for variables such as patient gender, number of recent hospitalizations, and creation to assignment interval, none showed a statistically significant association with absenteeism.

Data balanced

Synthetic minority over-sampling technique

SMOTE balances the data to increase the quantity of data from minority classes. This technique operates by generating synthetic data using similar neighboring samples and linear combinations among them, aiding the model in better understanding patterns of imbalanced classes (Rattan et al., 2021).

Adaptive synthetic sampling

Adaptive synthetic sampling (ADASYN) is a technique that, like SMOTE, addresses class imbalance when generating synthetic data. However, unlike SMOTE, ADASYN adapts the density of synthetic sample generation according to the classification difficulty of each instance. This adaptive approach allows greater attention to minority class regions that are more difficult for the model to learn (Alhudhaif, 2021).

Random split

Random split is used to perform random data splits, ensuring that both classes are represented proportionally in the training and testing sets. This approach helps mitigate the impact of class imbalance and facilitates the development of more balanced.

We selected the oversampling techniques SMOTE and ADASYN due to their clear advantages over conventional methods. Unlike random oversampling, which simply duplicates existing examples and can lead to overfitting, SMOTE and ADASYN generate new synthetic examples, increasing the representativeness of the minority class without duplication. Additionally, these techniques preserve the structure and complex relationships within clinical data, which is crucial for this type of dataset. ADASYN, in particular, excels in handling variability by generating more synthetic examples in areas where the minority class is more scarce, improving the model’s generalization capacity. This is especially relevant for complex machine learning models, which benefit from a more balanced and diverse dataset, outperforming conventional techniques that tend to lose valuable information or oversimplify the data (Dey & Pratap, 2023; Zhu et al., 2024). Data balancing was applied only to the training data, while the test data retained its original, unmodified structure. This approach ensures that the evaluation reflects the model’s performance on real, unbalanced datasets, providing a more accurate assessment of its practical applicability and robustness.

Data processing

Data normalization

It is an essential data preprocessing technique in machine learning used to standardize a dataset’s features. Its primary function is to adjust the features to a common scale by calculating each feature’s mean and standard deviation and then transforming the data to have a mean of 0 and a standard deviation of 1 (Aldi et al., 2023).

This work used the StandardScaler technique as a normalization technique, employing them as an experimental variation for the computational models.

Machine learning models

Decision tree: The decision tree classifier is a machine learning model for classifying and performing regression tasks. It is represented as a hierarchical tree structure, where each internal node makes decisions based on specific features, and the tree’s leaves represent predicted classes or values. Its ability to capture complex patterns and interpretability make it valuable, although it may be prone to overfitting. The construction process involves recursively splitting the data into subsets based on features to optimize the model’s accuracy (Charbuty & Abdulazeez, 2021). This model can be written as:

(1) f(x)=E[y|x]=∑m=1Mwm(x∈Rm)=∑m=1Mwmϕ(x;vm)

where Rm is defined as the m-th region. wm is a class label distribution for each leaf and vm represents the variable to be split. Finally ϕ is defined as ϕ(x)=[K(x,μ1),…,K(x,μN)] where μk are the complete training data or a given subset (Waldo-Benítez et al., 2024).

The Extra Trees Classifier (ETC) and the Random Forest Classifier (RFC) are machine learning models associated with decision trees used in classification and linear regression tasks. However, they need help with overfitting, resulting in issues when dealing with new data (Géron, 2022). To address this limitation, the RFC trains multiple decision trees randomly using subsets of the training data. We can select M unique trees for training from various subgroups, choosing them randomly and with replacement, to compute Eq. (2), where fm represents the m-th tree (Waldo-Benítez et al., 2024).

(2) f(x)=∑m=1M1Mfm(x).

On the other hand, the ETC introduces randomness in the training process to increase diversity among the trees, aiming to improve model performance (Rigatti, 2017). The Extra Tree Classifier’s default classification criterion is based on Gini impurity, as shown in Eq. (3), while it also offers entropy as an alternative criterion, as shown in Eq. (4): (3) GiniImpurity=∑j=1Ofj(1−fj)

(4) Entropy=∑j=1O−fjlog⁡(fj).

Fj is the frequency of label j at a node and O is the number of unique labels (Geurts, Ernst & Wehenkel, 2006).

Gradient boosting: Gradient boosting (GB) combines weak learning models into a single robust model (Friedman, 2002). The fundamental approach of Gradient Boosting’s training method is to start with a simple base model and adapt it to the training data. Subsequently, the residuals of the first base model are determined. A new weak model is trained using the residuals as the target in each subsequent iteration. This new model is added to the existing ensemble and adjusted to the updated residuals (Géron, 2022). This model can be expressed as follows:

(5) Y=M(x)+error

Y represents the dependent variable, while M(x) denotes the decision tree based on the independent variables x. Now, the error is predicted from the previous decision tree:

(6) error=G(x)+error2

Therefore, as an example, two decision trees are considered (Ayyadevara, 2018).

Support vector machine (SVM): This model can address both classification and regression problems, whether linear or non-linear. It is particularly suitable for small to moderately complex datasets. SVM classification separates classes by maximizing decision boundaries concerning the closest training patterns. Furthermore, it aims to maximize the distance from the most relevant training pattern by introducing non-linearity. SVM achieves linear class separation using kernel functions that modify or add features based on the training set (Géron, 2022). The SVM algorithm is expressed as:

(7) MaxL(α)=∑i=1nαi−12∑i=1n∑j=1nyiyjαiαjK(xi,xj)s.t.C≥α≥0;∀i=1,...,n∑i=1nyiαi=0

where α are the Lagrange multipliers, C is a penalty hyperparameter, xi, zj ∈Rd are two given training vectors; and K(x,z) in Rd × Rd kernel function K(x,z)=(α(x),α(z)) (Waldo-Benítez et al., 2024).

Bagging random forest: Bagging (Bootstrap Aggregating) is an ensemble modeling technique that involves creating multiple training datasets through sampling with replacement. Each set is used to train base models, and the predictions of these models are combined to obtain a more robust final prediction (Kadiyala & Kumar, 2018). RandomForest combines the Bagging technique with the introduction of randomness in tree construction to create a more robust and accurate ensemble of trees. The Bagging model can be defined as:

(8) θ^(x)=argminθ(x)∈Θ⁡L(θ(x))

In this context, Θ represent a function class that can be represented by the estimator, such as neural networks or decision trees. The objective function L( θ(x)) is an estimate based on the data of the expected value of some functional, such as the negative log-likelihood or another loss function. ‘Bagging’ consists of repeatedly drawing random resamples xb of the data, and then either optimizing the value of L( θ( xb)) averaged over the resamples, or averaging the resampled values of θ^ (Bühlmann & Yu, 2002).

AdaBoost decision tree: A machine learning algorithm combines the AdaBoost technique with decision trees. AdaBoost assigns weights to training instances based on their classification difficulty and sequentially trains decision trees to correct previous errors, thus improving the overall model accuracy. This weighted ensemble approach enables the model to adapt effectively to complex and imbalanced datasets (Ampomah et al., 2021). For the formal description of AdaBoost, let the training set be Dn = ( x1, y1), …, ( xn, yn). The algorithm runs for T iterations. T is the only pre-specified hyper-parameter of AdaBoost that can be set, for example, by cross-validation. In each iteration t = 1, …,T, we choose a base classifier h(t) from a set H of classifiers and set its coefficient α(t). The output of AdaBoost is a discriminant function constructed as a weighted vote of the base classifiers (Cao et al., 2013).

(9) f(T)(⋅)=Δ⁡∑t=1Tα(t)h(t)(⋅).

Hyperparameters

Grid search

Grid search is a commonly employed technique in hyperparameter optimization for machine learning models. This strategy involves exploring a specific set within the hyperparameter space of a model. However, the main limitation of this approach lies in the dimensionality of the space, requiring constraints in specific areas as the hyperparameters of each model are considered independent (Jiang & Xu, 2022). For this work, grid search was applied to each computational model with the following combinations, as shown in Table 3.

Table 3 Combination in the database preprocessing for the GridSearch.

Database combination	Preprocessing	
1	Unbalanced database	
2	StandardScaler	
3	RandomSplit	
4	RandomSplit, StandardScaler	
5	SMOTE	
6	SMOTE, StandardScaler	
7	ADASYN	
8	ADASYN, StandardScaler	

Table 4 shows the parameters to be optimized for each computational model.

Table 4 Hyperparameter optimization in each of the models.

Model	Hyperparameter	
Random Forest	n_estimators	
	criterion	
	max_depth	
	min_samples_split	
	min_samples_leaf	
	max_features	
Decision Tree	criterion	
	max_depth	
	min_samples_split	
	min_samples_leaf	
	max_features	
	min_impurity_decrease	
	splitter	
	cc_alpha	
Bagging Random Forest With optimized hyperparameters by RandomForest	criterion	
	max_depth	
	min_samples_split	
	min_samples_leaf	
	max_features	
	min_impurity_decrease	
	splitter	
	cc_alpha	
AdaBoost Decision Tree With optimized hyperparameters by DecisionTree	n_estimators	
	learning_rate	
	algorithm	
	random_state	
Extra Trees	n_estimators	
	criterion	
	max_depth	
	min_samples_split	
	min_samples_leaf	
	max_features	
	bootstrap	
	random_state	
Gradient Boosting	n_estimators	
	learning_rate	
	max_depth	
SVM	C	
	kernel	
	degree	
	gamma	
	tol	

Metrics

The metrics in computational models are essential for evaluating their performance, focusing on false positives, false negatives, true positives, and true negatives. These measures quantify the accuracy and predictive ability of the model, allowing adjustments to enhance its performance (Carvalho, Pereira & Cardoso, 2019). The following are the most fundamental metrics.

Accuracy

Accuracy is used as an indicator to assess the effectiveness of a classification model in its predictive capabilities. Its calculation involves dividing the number of correct predictions the model makes by the total number of predictions generated. This accuracy value is expressed as a fraction ranging from 0 to 1, representing the percentage of correct predictions made by the model (Mora-Rubio et al., 2023).

(10) Accuracy=TP+TNTP+TN+FP+FN.

Precision

The precision metric evaluates the proportion of positive cases a model correctly identifies relative to the total number of cases identified as positive, encompassing both true and false positives (Liu, Li & Li, 2020).

(11) Precision=TPTP+FP.

F1 score

F1 score is a crucial metric for evaluating a model’s accuracy in identifying positive and negative cases, especially in scenarios with imbalanced data. Calculated as the harmonic mean of precision and recall, it is ideal in situations with an uneven class distribution (Battista, Salvatore & Castiglioni, 2017).

(12) F1=2∗Precision∗RecallPrecision+Recall.

Confusion matrix

A confusion matrix is a machine learning tool that summarizes a classification model’s performance by showing the number of correct and incorrect predictions in terms of positive and negative classes. The matrix consists of four components: true positives (TP), false positives (FP), true negatives (TN), and false negatives (FN) (Heydarian, Doyle & Samavi, 2022).

Cross validation

A model’s performance evaluation is carried out using cross-validation (CV), which involves dividing the dataset into smaller subsets called ‘folds’ (k) of comparable sizes. This process generates a series of iterations in which the model is trained and evaluated, then an average is calculated (Rushing et al., 2015). The mathematical representation of this procedure is expressed by the following equation:

(13) Cross-Validation=1k∑i=1kPerformancei.

In this work, a cross-validation with k = 5 was implemented for each computational model and their combination.

Feature importance

Evaluation of feature importance is essential to interpret machine learning models. It allows an understanding of their functioning and identification of biases and critical features. This technique is crucial in an environment where artificial intelligence models are becoming increasingly complex and challenging to interpret due to scientific advancements (Chen et al., 2020). To evaluate feature importance in this study, the models were trained using hyperparameters optimized through grid search, selecting the combination that maximized accuracy in cross-validation. Subsequently, the feature importances attribute from scikit-learn was used to measure the contribution of each feature by calculating the cumulative decrease in impurities, based on the reduction of entropy or Gini index during the models’ splits, thereby reflecting each feature’s influence on predictions (Paper, 2020).

SHAP values

SHAP values are a game theory-based technique to interpret machine learning models. In this study, we used the SHAP Python library to calculate SHAP values and assess the influence of each feature on the model’s predictions (Victoria, Tiwari & Ghulam, 2024). The calculation of SHAP values considers how the model’s prediction changes when a feature is included or excluded, providing a precise measure of its impact. This tool allows us to identify the most critical features and analyze the stability and robustness of the evaluated models. The main objective of using SHAP values is to complement the quantitative metrics analysis, providing a solid statistical basis to determine whether a model is significantly more stable and robust than others. This approach also enhances the interpretability of the results, which is crucial in contexts where transparency is essential (Marcílio & Eler, 2020).

Evaluation method

The study evaluated the model’s performance using key metrics to ensure balance and accuracy in classification. The final decision was based on the interpretability provided by SHAP values, identifying the most relevant features for predictions. This approach ensures reliable and applicable results in the healthcare domain.

Hardware and resources

The exploratory data analysis and implementation of computational models were carried out using Visual Studio Code with Python 3.10.0, an NVIDIA Quadro M2000 graphics card, and 32 GB of RAM. The statistical analysis to identify significant patterns and relationships was conducted in RStudio 4.2.3.

Human studies declaration

The bioethics research committee of the Fundación Valle del Lili approved the academic protocol under institutional approval act No. 281. The study does not require informed consent.

Results

Database preprocessing

Data balanced

To balance the distribution of classes in the dataset, three balancing techniques were implemented: RandomSplit, SMOTE, and ADASYN. This process generated three additional datasets. In the first dataset, created with RandomSplit, the number of instances of the Attendances class was randomly reduced to match that of the Absences class. Additionally, the minority class was increased using SMOTE and ADASYN (Fig. 2).

Figure 2 Data balanced by methods.

We conducted additional analyses using three datasets generated with Random Split, ADASYN, and SMOTE techniques to compare the distributions of patient characteristics with the original unbalanced dataset (Table 2). The corresponding summary tables are provided in Supplemental Information S1. While the distributions of demographic and clinical characteristics were generally comparable, the statistical significance of variables such as sex (original p = 0.2) and recent hospitalizations (original p = 0.5) changed in the balanced datasets. These changes suggest that the balancing methods could influence the statistical properties of the data. Therefore, these modifications should be carefully considered when interpreting the results and their generalizability to real clinical settings.

Correlation matrix

In Fig. 3, the correlation matrix of the dataset is presented, revealing the lack of linear correlation between the features and suggesting independence among them. This positive independence is observed when analyzing attributes that are expected not to be directly related. The correlation values were calculated using the Pearson correlation coefficient through the corr() method in Python (Molin, 2019). For numeric features, the Pearson coefficient measures the linear relationship between variables, ranging from −1 to 1. For multicategorical variables, these were first encoded into numerical representations before calculating the correlation, allowing their inclusion in the analysis. The low correlation indicates less-defined patterns, benefiting the model’s adaptability to the complexity of the dataset. Moreover, the lack of correlation helps prevent overfitting during model training by avoiding the capture of false relationships, thereby enhancing the model’s ability to generalize more effectively to new datasets (Nasution, Sitompul & Ramli, 2018).

Figure 3 Correlation matrix.

Models

Hyperparameters

Hyperparameter optimization was applied for each combination concerning the database preprocessing, as specified in section Hyperparameters. Table S1 shows the results for each of the classes in these combinations.

Model metrics for each preprocessing method

The machine learning models were evaluated using the hyperparameters derived in the gridsearch process using metrics such as accuracy, recall, and F1-score, as well as both standardized and non-standardized data. Table 5 presents the evaluation of the models with the unbalanced database; Table 6 shows their metrics using the RandomSplit balancing technique; Table 7 presents the model metrics using the ADASYN balancing technique; and Table 8 displays the results with the SMOTE balancing technique. The data normalization process was carried out using the StandardScaler technique.

Table 5 Results obtained for each model evaluated in two classes with the unprocessed database include the accuracy obtained in the 5-fold cross-validation, the F1 score, and the recall.

In addition, each result is presented for normalized (using the StandardScaler technique) and non-normalized data.

Unprocessed database	
Model	Standard scaler	Non standard scaler	
	Accuracy cross-validation (%)	F1-score (%)	Recall (%)	Accuracy cross-validation (%)	F1-score (%)	Recall (%)	
Decision Tree	79.08 ± 0.51	84.33	87.87	79.06 ± 0.53	84.33	87.87	
Random Forest	81.09 ± 0.39	86.16	89.62	81.12 ± 0.54	86.58	90.35	
AdaBoost Decision Tree	81.08 ± 0.54	85.98	88.76	81.07 ± 0.54	85.96	88.76	
Bagging Random Forest	81.44 ± 0.57	86.49	91.37	81.29 ± 0.57	86.66	91.82	
Gradient Boosting	82.02 ± 0.31	87.35	90.15	82.06 ± 0.26	87.37	90.15	
Extra Trees	81.25 ± 0.9	86.95	91.33	81.25 ± 0.9	86.95	91.33	
SVM	76.83 ± 0.62	84.14	88.64	73.47 ± 1.25	82.25	91.04	

Table 6 Results obtained for each model evaluated in two classes with balancing by RandomSplit include the accuracy obtained in the 5-fold cross-validation, the F1 score, and the recall.

In addition, each result is presented for normalized (using the StandardScaler technique) and non-normalized data.

Random split balanced	
Model	Standard scaler	Non standard scaler	
	Accuracy cross-validation (%)	F1-score (%)	Recall (%)	Accuracy cross-validation (%)	F1-score (%)	Recall (%)	
Decision Tree	75.9 ± 1.09	76.37	74.45	76.02 ± 0.99	76.26	74.22	
Random Forest	79.12 ± 1.07	78.67	76.74	78.94 ± 1.02	78.90	76.51	
AdaBoost Decision Tree	79.06 ± 1.06	78.16	76.82	79.08 ± 1.21	77.70	76.66	
Bagging Random Forest	80.12 ± 1.01	79.42	77.81	79.74 ± 0.87	79.27	77.66	
Gradient Boosting	80 ± 1.29	79.88	80.18	80 ± 1.3	79.68	79.65	
Extra Trees	79.14 ± 1.03	79.13	76.89	79.14 ± 1.03	79.13	76.89	
SVM	73.95 ± 1.47	75	74.83	71.74 ± 0.56	72.07	71.77	

Table 7 Results obtained for each model evaluated in two classes with synthetic balancing by ADASYN include the accuracy obtained in the 5-fold cross-validation, the F1 score, and the recall.

In addition, each result is presented for both normalized (using the StandardScaler technique) and non-normalized data.

ADASYN balanced	
Model	Standard scaler	Non standard scaler	
	Accuracy cross-validation (%)	F1-score (%)	Recall (%)	Accuracy cross-validation (%)	F1-score (%)	Recall (%)	
Decision Tree	76.07 ± 0.31	78.63	75	76.33 ± 0.2	78.95	75.73	
Random Forest	82.12 ± 0.81	83.07	79.23	82.29 ± 0.91	82.66	78.99	
AdaBoost Decision Tree	84.23 ± 0.58	84.67	83.43	76.97 ± 0.49	79.49	75.73	
Bagging Random Forest	83.53 ± 0.39	84.61	82.17	83.9 ± 0.44	84.01	80.86	
Gradient Boosting	82.41 ± 0.86	81.07	84.80	82.39 ± 0.84	84.79	81.03	
Extra Trees	82.38 ± 0.67	83.26	81.19	82.38 ± 0.67	83.26	81.19	
SVM	76.56 ± 0.64	79.68	74.06	82.37 ± 0.28	78.75	84.73	

Table 8 Results obtained for each model evaluated in two classes with synthetic balancing by SMOTE include the accuracy obtained in the 5-fold cross-validation, the F1 score, and the recall.

In addition, each result is presented for both normalized (using the StandardScaler technique) and non-normalized data.

SMOTE balanced	
Model	Standard scaler	Non standard scaler	
	Accuracy cross-validation (%)	F1-score (%)	Recall (%)	Accuracy cross-validation (%)	F1-score (%)	Recall (%)	
Decision Tree	77.31±0.59	78.83	70.81	77.34±0.62	78.7	70.64	
Random Forest	82.93±0.73	83.41	80.94	82.93±0.62	83.27	80.66	
AdaBoost Decision Tree	84.39±0.59	84.22	84	84.41±0.65	84.23	83.96	
Bagging Random Forest	84.80±0.81	84.75	83.02	84.79±1.03	84.3	81.76	
Gradient Boosting	83.14±0.84	85.12	81.96	83.23±0.84	85.11	81.92	
Extra Trees	83.17±0.66	83.6	82.21	83.17±0.62	83.6	82.21	
SVM	77.76±0.54	80.44	75.45	77.61±0.47	78.29	74.88	

For better interpretability of the results, three computational methods (AdaBoost, Bagging, and GradientBoosting) were selected due to their relevant metric values and their stability in the relationship between them. The F1-score metric, especially relevant in scenarios with imbalanced classes, was analyzed. Figure 4 compares the F1-score across different balancing approaches and shows that, in general, the variations between them are minimal. However, the RandomSplit undersampling method shows lower F1-score performance for AdaBoost and Bagging. Since the metrics do not show significant differences, interpretability was prioritized to select the most appropriate model, valuing its ability to provide clear and useful insights in this context.

Figure 4 F1-score comparison.

SHAP values highlight that the most influential variables in predicting medical appointment no-shows include the number of previous no-shows, the month and day of the appointment, the number of prescribed medications, and diagnosed diseases, among others. Gradient Boosting was discarded due to the instability and range of its SHAP values. At the same time, AdaBoost showed greater sensitivity to the variable “Number of Previous Attendance,” whose relatively higher impact could lead to erroneous interpretations. In contrast, Bagging demonstrated a more uniform and stable distribution of SHAP values, emphasizing key variables’ significant and consistent impact. These characteristics set Bagging apart, establishing it as the most reliable and aligned option with the study’s objectives. These results can be observed in the Supplemental Information S2.

Based on the metrics obtained (accuracy, F1-score, and recall) presented in Tables 5, 7 and 8, and the distribution of SHAP values (Fig. 5), SMOTE was selected as the best balancing strategy for the Bagging model. Specifically, SMOTE (Fig. 5A) provides a more uniform and consistent distribution of key variables, such as the number of previous no-shows, the day, the month, and the number of medications. These variables reflect a behavior most closely aligned with the phenomenon of no-shows, reinforcing the model’s reliability. In contrast, with ADASYN (Fig. 5B), variables like the day and time exhibited greater dispersion, while in the unbalanced dataset, the relevant variables showed a less defined impact. Therefore, SMOTE not only improved performance metrics but also ensures a more stable and accurate representation of the factors associated with medical no-shows, solidifying its choice as the optimal strategy for this context.

Figure 5 SHAP values for Bagging RandomForest with standarized data: (A) SMOTE balancing (B) ADASYN balancing, and (C) unbalanced data.

Figure 6 shows the results obtained with the Bagging RandomForest model with the standardized and SMOTE-balanced data. It includes the confusion matrix for the two classes and their respective AUC values for each combination and class type in the standardized data preprocessing.

Figure 6 Bagging RandomForest confusion matrix, and ROC curve.

Analysis of the confusion matrix reveals adequate discrimination of the class of interest. This differentiation capacity correlates positively with the behavior observed in the ROC curve, whose value (0.89) demonstrates a good class discrimination capacity. Supplemental Information S3 contains the class-wise metric reports of the Bagging RandomForest model for each data preprocessing combination, as well as the feature importances for each of them.

Figure 7 illustrates the feature importance of the Bagging RandomForest model, which align with the impact of the variables presented with the SHAP values.

Figure 7 Feature importance plot obtained from the Bagging RandomForest model with SMOTE.

Discussion

In our study, we evaluated seven computational models to address the issue of medical absenteeism in a dataset with discrete and categorical features. The models analyzed were random forest, decision tree, Bagging, AdaBoost, ExtraTrees, GradientBoosting, and SVM, whose results are presented in Tables 5, 6, 7 and 8. The results revealed that ensemble models such as Bagging, AdaBoost, ExtraTrees, and GradientBoosting outperformed simpler models like random forest and decision tree. We highlighted the metrics obtained by Bagging RandomForest, which demonstrated high precision, sensitivity, and specificity. This model excels in efficiently handling both continuous and categorical features and its robustness against class imbalance, making it a solid option for predicting medical absenteeism.

The Bagging RandomForest model uses the Bagging ensemble technique to build multiple independent decision trees from random subsets of the dataset. Each tree generates a prediction, and the model combines these predictions to produce a more robust final result. This approach reduces variance, improves generalization capacity, and decreases the risk of overfitting, especially in scenarios with imbalanced data. Furthermore, its ability to handle mixed data types (categorical and continuous) makes it particularly effective for complex problems like medical absenteeism.

Our analysis revealed that when the SMOTE balancing technique is combined with standardized data, the Bagging RandomForest model showed superior performance compared to other techniques for class 0 (no-shows). This finding was reflected in an improvement in evaluation metrics, as shown in Fig. 6 and Table 8. The effectiveness of the SMOTE technique lies in its ability to generate synthetic samples uniformly distributed within the feature space, preserving the structure of the minority class data. This is especially relevant given the nature of our dataset. Moreover, the introduction of greater diversity in the dataset improved the model’s generalization capacity. The inclusion of the datasets generated with Random Split, ADASYN, and SMOTE allowed us to evaluate the robustness of our findings under different balancing methods. While these techniques improved the representation of the minority class, they also altered the statistical significance of certain variables, such as sex and recent hospitalizations, which were not significant in the original unbalanced dataset (Table 2). This highlights their impact on the relationships between variables and outcomes. Despite these modifications, the consistency of predictive performance and the robustness of Bagging RandomForest as the selected model reinforce the reliability of the approach. However, it is crucial to interpret statistical associations derived from synthetic datasets with caution.

To enhance interpretability in the selection of computational models, SHAP values were applied. This technique allows for the evaluation of the individual impact of each variable on the model’s predictions. This analysis, complemented by the evaluation of feature importance from the computational model, confirmed the performance of the Bagging RandomForest model. The results showed that the impact of the variables in this model aligns more closely with the real behavior of medical absenteeism, reinforcing its relevance and reliability as a predictive tool in this context. Identifying the most impactful variables through SHAP values in the Bagging model, complemented by a statistical analysis of p-values that align with these results (Table 2), enables data-driven decision-making and provides valuable insights into the key factors in predicting medical appointment no-shows. Variables such as the number of previous no-shows, the day and month of the appointment, and diagnosed diseases emerge as critical determinants (Fig. 5A). These findings have significant practical implications, as they enable a detailed analysis of behavioral patterns associated with absenteeism and offer insights into the factors that determine this phenomenon. This knowledge facilitates the development of personalized strategies, such as reminders adjusted to days with higher absenteeism rates, appointment schedules tailored to identified patterns, and prioritization of follow-up for patients with a history of multiple absences. By focusing efforts on patients most likely to miss appointments, these interventions not only improve operational efficiency but also increase the effectiveness of resource allocation. Furthermore, these insights allow healthcare institutions to proactively address absenteeism, optimizing the patient experience through timely, personalized care and ensuring more efficient management of medical resources. This integration of data-driven insights into strategic planning ensures that interventions are targeted where they are most needed, enhancing overall healthcare outcomes.

Furthermore, leveraging these insights through a computational model supports decision-making and serves as a practical tool for the institution. This model strengthens the ability of institutions to act proactively against medical absenteeism, offering a reliable predictive tool that facilitates data-driven decision-making and optimizes strategic planning. By effectively segmenting the population based on the risk of absenteeism, institutions can focus resources where they are most needed, mitigating operational inefficiencies and improving the patient experience by ensuring more timely and personalized care. This integration of data-driven insights and strategic planning ensures that resources are directed toward critical areas, enhancing interventions’ effectiveness and improving healthcare system outcomes.

Conclusion

This comparative study evaluated machine learning models to classify medical absenteeism using various data encompassing clinical, sociodemographic, and specific characteristics of the patient’s latest medical appointment in the context of internal medicine. The validity of the variables in the structured database was examined by identifying statistical significance values (p and q), revealing the significant influence of several variables. Among these, patient age, type of medical insurance, number of diagnosed diseases, recent hospitalization history, the time, day, and month of appointment scheduling, and the number of previous absences stood out. These findings underscore the importance of various factors in predicting medical absenteeism, providing a more comprehensive understanding of the determinants influencing this phenomenon. Ensemble models outperformed traditional ones in precision and stability, with Bagging RandomForest, GradientBoosting, ExtraTrees, and AdaBoost Decision Tree standing out. Bagging RandomForest proved especially effective in identifying patients at risk of missing their appointments, even with imbalanced or balanced data using techniques such as RandomSplit, SMOTE, and ADASYN. The SMOTE technique allowed the generation of synthetic samples uniformly distributed, preserving the structure of the minority class data and reducing bias toward the majority class.

Despite the low correlation between variables, computational models were able to identify complex patterns and nonlinear relationships. Data standardization also improved accuracy by unifying categorical and discrete features, enabling more efficient processing and greater prediction accuracy. The methodology of the Bagging RandomForest model, which integrates SMOTE and data standardization, achieved a cross-validation accuracy of 84.80 ± 0.81% and an F1-score of 83.02%, confirming its effectiveness in predicting medical absenteeism in a clinical setting. Additionally, the SHAP values of the model helped identify the most impactful variables in predictions, aligning closely with the natural behavior of absenteeism. Variables such as the number of previous absences, the day and month of the appointment, the number of diagnosed diseases, and prescribed medications emerged as key determinants in the model’s predictions. This approach not only enhanced the model’s interpretability but also provided a solid foundation for developing personalized strategies targeted at the population of interest. For instance, by identifying patients with multiple absenteeism histories, medical institutions can implement personalized reminders, adjust appointment schedules based on specific patterns, or prioritize follow-ups for patients with complex conditions. These strategies allow resources to be allocated efficiently, maximizing intervention effectiveness and improving attendance rates. Knowledge of impactful variables strengthens the ability to make informed, data-driven decisions, adapting actions to real-world contexts and ensuring a more precise and effective approach to appointment management.

This study successfully identified a robust and efficient computational model for predicting medical absenteeism, opening opportunities for future research. It is recommended to include broader sociodemographic characteristics of patients, as these could capture regional behavioral patterns and provide a more complete perspective on the factors influencing absenteeism. Furthermore, scaling the model to other medical specialties would require analyzing the specific behaviors of these specialties and determining relevant variables of interest, as the determinants of absenteeism can vary significantly across clinical contexts. A computational model that identifies patients at higher risk of missing appointments represents a valuable tool for institutional support. This approach not only complements existing strategies to mitigate absenteeism but also helps optimize resource management, personalize interventions, and improve operational efficiency in healthcare systems. These future directions will not only strengthen the model’s applicability but also expand its impact on improving healthcare outcomes.

Supplemental Information

Supplemental Information 1 Statistical description of the data generated by the balancing methods.

Tables of statistical description of the data generated by the balancing methods

Supplemental Information 2 Shap values in computational models.

Figures of the shap values for the AdaBoost, Bagging and GradientBoosting models for each data preprocessing.

Supplemental Information 3 Bagging RandomForest model metrics.

Bagging RandomForest model metrics per data preprocessing

The manuscript was grammatically revised and corrected using the artificial intelligence tool ChatGPT-4.

Additional Information and Declarations

Competing Interests

The authors declare that they have no competing interests.

Author Contributions

Felipe Ocampo Osorio conceived and designed the experiments, performed the experiments, analyzed the data, performed the computation work, prepared figures and/or tables, authored or reviewed drafts of the article, and approved the final draft.

Santiago Pedroza Gomez conceived and designed the experiments, analyzed the data, prepared figures and/or tables, and approved the final draft.

David Esteban Rebellón Sanchez performed the experiments, analyzed the data, prepared figures and/or tables, and approved the final draft.

Richard Ramirez Fernandez conceived and designed the experiments, analyzed the data, prepared figures and/or tables, and approved the final draft.

Reinel Tabares-Soto performed the computation work, authored or reviewed drafts of the article, and approved the final draft.

Mario Alejandro Bravo-Ortíz conceived and designed the experiments, performed the experiments, analyzed the data, performed the computation work, prepared figures and/or tables, authored or reviewed drafts of the article, and approved the final draft.

Gustavo Adolfo Cruz Suarez conceived and designed the experiments, performed the experiments, analyzed the data, authored or reviewed drafts of the article, and approved the final draft.

Ethics

The following information was supplied relating to ethical approvals (i.e., approving body and any reference numbers):

Fundación Valle del Lili, Comité de Ética en Investigación Biomédica, No. 281.

Data Availability

The following information was supplied regarding data availability:

The dataset and code to reproduce the analyses and scripts are available at Zenodo: Unidad de Inteligencia Artificial-Fundación Valle del Lili. (2025). UIA-FVL/Non-ShowsML: v1.0.0 (DOI). Zenodo. https://doi.org/10.5281/zenodo.14645465.

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
