# Peer review of "Predicting no-shows at outpatient appointments in internal medicine using machine learning models"

_PeerJ Computer Science, doi:10.7717/peerj-cs.2762_

## Round 0.1 · original submission · Major Revisions

Thank you for submitting the manuscript to PeerJ Computer Science. After careful consideration, we would like to invite you to submit a revision of the manuscript to address all comments and concerns raised by the reviewers.

Reviewer 1 ·

Basic reporting

Osorio et al. build machine learning models to predict non-shows at outpatient in internal medicine at one hospital center in Columbia. The aim of this work is important in improving patient care and efficiency in medical resource. Using the selected features from electronic records of ~18k patients from 2019-2022, the authors evaluated the prediction accuracy for non-shows between several decision trees, boosting methods and SVM, as well as combined with oversampling approach such as ADASYN and SMOTE. The authors found GradientBoosting performs the best with high cross-validation accuracy as ~82%. The method section of this manuscript is written well, while the results need more clarifications.

Experimental design

1. For the 3 additional datasets generated, to make sure they can be comparable to the raw observed data, can the authors provide a patient characteristics table for each features like Table 2?
2. Figure 1(b) shows distribution of age, but the title and caption says “Gender”. Please make it consistent. In line 145-146, the authors’ conclusion looks not right. Figure 1(b) shows elder people have lower rate of absence, as indicated by the numbers in Table 2.
3. Can the authors describe how correlation is computed for the correlation matrix in Figure 3, especially for multi-class variables?
4. Please describe method/metrics to evaluate feature importance in the “Feature importance” method section.

Validity of the findings

1. Please make sure number are separated by commas instead of dots, such as in line 138-140.
2. For results from Table 5-8, can the authors make figures for these metrics to make it easier for comparison? Also, please make tables in orders.
3. Regarding the conclusions made from line 295-300, the authors said GradientBoosting is more stable and robust than other methods. Can the authors give a statistical description on this? Are they significantly different?

Reviewer 2 ·

Basic reporting

As stated in the introduction, the study aims to predict no-show risks using clinical and sociodemographic characteristics. However, the dataset appears to lack meaningful representation of sociodemographic factors. For instance, no-show behaviors might correlate with personality traits or demographic differences, as different groups may exhibit varying attitudes toward medical appointments.
o Consider revisiting the dataset to include features that better capture relevant sociodemographic characteristics.
o Discuss or explore how these features influence the decision-making process, as this could provide valuable insights and make the findings more impactful.
These refinements will not only enhance the study’s methodological rigor but also increase its practical significance and contribution to the field.

Experimental design

2. Feature Relevance and Insights:
The dataset includes 12 features, as highlighted in Figure 5’s correlation matrix and feature importance analysis. However, some features, like “appointment month”, seem tenuously related to predicting no-shows. While reporting model accuracy is important, it is equally critical to explain the model’s decision-making process and extract actionable insights. This aspect is currently underdeveloped in the manuscript and represents a significant gap.

Validity of the findings

The reported accuracy of 82.41% suggests moderate predictive performance, with a random guess baseline at 50%. However, the reliability of this model for practical applications is questionable, given that the error rate implies a 1 in 5 chance of incorrect predictions. To improve robustness, consider the following:
o Apply feature pruning strategies to remove irrelevant or noisy features.
o Explore alternative models, such as Convolutional Neural Networks (CNNs) or other advanced machine learning techniques, which may better capture complex patterns in the data.

Additional comments

NA

Annotated reviews are not available for download in order to protect the identity of reviewers who chose to remain anonymous.

Reviewer 3 ·

Basic reporting

This paper explores the use of various machine learning techniques to predict patient absenteeism at outpatient appointments.
1. The manuscript generally uses clear and professional English. However, some sections could benefit from less passive voice to enhance clarity and engagement.
2. The paper adequately references relevant literature, providing a solid theoretical foundation for the study.
3. The paper claims that a specific model exhibited the best performance, yet this assertion is not supported by the data shown in the tables and figures.

Experimental design

1. The conclusion suggesting that the ADASYN balancing technique provided superior discrimination of the target class may be misleading. Since ADASYN generates synthetic data points, the model's performance as evaluated on these artificially balanced datasets might not accurately reflect its effectiveness on real, unbalanced data. Ideally, the model should be trained on balanced data but evaluated on the original, unmodified dataset to truly assess its practical applicability and robustness.
2. Table 5 is followed by Table 8.
3. In Figure 4, all four models showed similar ROC of 0.89 to 0.9. And the unbalanced data had ROC of 0.9. Which does not support the claim of choosing ADYSYN as the final model.
4. In Table 5, the Unprocessed data had highest F1-Score of 87.37 (Non-standardScalar). I understand that the author prioritized using cross-validated accuracy. But in the case of comparison of balanced and unbalanced data, I think F1 score should be a better metric. Additionally, while there is <1% difference in accuracy among [Unprocessed data, Non-standardScalar, and gradiant boosting] & [ADYSYN, Standardize Scalar, gradiant boosting], there is almost a 6% increase in F1. This finding does not support the choice of standardization or applying data balancing techniques.

Validity of the findings

1. The paper could benefit from more explicitly assessment of the impact and novelty of its findings.
2. The conclusions are well-articulated regarding the immediate results but fail to adequately address broader implications, unresolved questions, or future research directions.

Additional comments

No comments.

---

## Round 0.2 · accepted · Accept

Thank you for submitting the article to PeerJ Computer Science. All reviewers have no further suggestions and agree to accept the article for publication.

Reviewer 1 ·

Basic reporting

The authors addressed all my comments

Experimental design

NA

Validity of the findings

NA